# A diuranium carbide cluster stabilized inside a $C_{80}$ fullerene cage

Xingxing Zhang[1], Wanlu Li[2], Lai Feng[3], Xin Chen[2], Andreas Hansen[4], Stefan Grimme[4], Skye Fortier[5], Dumitru-Claudiu Sergentu[6], Thomas J. Duignan[6], Jochen Autschbach [6], Shuao Wang[7], Yaofeng Wang[1], Giorgios Velkos[8], Alexey A. Popov [8], Nabi Aghdassi[9], Steffen Duhm [9], Xiaohong Li[1], Jun Li[2], Luis Echegoyen[4], W.H.Eugen Schwarz [2,10] & Ning Chen[1]

Unsupported non-bridged uranium–carbon double bonds have long been sought after in actinide chemistry as fundamental synthetic targets in the study of actinide-ligand multiple bonding. Here we report that, utilizing $I_h(7)$-$C_{80}$ fullerenes as nanocontainers, a diuranium carbide cluster, U=C=U, has been encapsulated and stabilized in the form of UCU@$I_h(7)$-$C_{80}$. This endohedral fullerene was prepared utilizing the Krätschmer–Huffman arc discharge method, and was then co-crystallized with nickel(II) octaethylporphyrin (Ni$^{II}$-OEP) to produce UCU@$I_h(7)$-$C_{80}$·[Ni$^{II}$-OEP] as single crystals. X-ray diffraction analysis reveals a cage-stabilized, carbide-bridged, bent UCU cluster with unexpectedly short uranium–carbon distances (2.03 Å) indicative of covalent U=C double-bond character. The quantum-chemical results suggest that both U atoms in the UCU unit have formal oxidation state of +5. The structural features of UCU@$I_h(7)$-$C_{80}$ and the covalent nature of the U($f^1$)=C double bonds were further affirmed through various spectroscopic and theoretical analyses.

[1] Laboratory of Advanced Optoelectronic Materials, College of Chemistry, Chemical Engineering and Materials Science, Soochow University, Suzhou, Jiangsu 215123, China. [2] Department of Chemistry and Key Laboratory of Organic Optoelectronics & Molecular Engineering of the Ministry of Education, Tsinghua University, Beijing 100084, China. [3] Soochow Institute for Energy and Materials InnovationS (SIEMIS), College of Physics, Optoelectronics and Energy & Collaborative, Soochow University, Suzhou, Jiangsu 215006, China. [4] Mulliken Center for Theoretical Chemistry, Universität Bonn, 53115 Bonn, Germany. [5] Department of Chemistry, University of Texas at El Paso, 500 West University Avenue, El Paso, TX, 79968, USA. [6] Department of Chemistry, University at Buffalo, State University of New York, Buffalo, NY 14260-3000, USA. [7] School of Radiological and Interdisciplinary Sciences & Collaborative Innovation Center of Radiation Medicine of Jiangsu, Higher Education Institutions, Soochow University, Suzhou, Jiangsu 215123, China. [8] Nanoscale Chemistry, Leibniz Institute for Solid State and Materials Research, 01069 Dresden, Germany. [9] Institute of Functional Nano & Soft Materials (FUNSOM), Soochow University, Suzhou, Jiangsu 215123, China. [10] Physikalische und Theoretische Chemie, Universität Siegen, 57068 Siegen, Germany. These authors contributed equally: Xingxing Zhang, Wan-lu Li, Lai Feng. Correspondence and requests for materials should be addressed to J.L. (email: junli@tsinghua.edu.cn) or to L.E. (email: echegoyen@utep.edu) or to N.C. (email: chenning@suda.edu.cn)

Understanding the nature of actinide-ligand multiple bonding remains a modern day challenge owing to the complex electronic structures of these elements, and as a consequence, our comprehension of their chemistries lags behind that of the more commonly studied transition metals[1–5]. Importantly, recent developments in the synthesis and study of molecular uranium complexes containing a variety of U=E(L) or U≡E(L) moieties (E = O, S, Se, Te, N, NR, P, As; L = other ligands) have provided valuable insights regarding the participation of the 5f and 6d valence orbitals in chemical bonding[1,2,6–27]. Conspicuously missing from this list are U=C bonds, specifically those which lack ancillary heteroatom or chelating support. While metal–carbon double bonds (i.e., Schrock carbenes) are common in transition metal chemistry and catalysis, unsupported U=C bonds have long remained a major and outstanding synthetic target. Some success towards the synthesis of An=C bonds has been realized through the use of heteroatom stabilizing chelating ligands such as U=[C(Ph$_2$PNSiMe$_3$)$_2$](O)Cl$_2$ or U=[C(Ph$_2$PS)] (BH$_4$)$_2$ (THF)$_2$, among others[4,27–33]. In a few rare instances, the isolation of non-chelated U=C bonds has been achieved in complexes such as ($\eta^5$-C$_5$H$_5$)$_3$U=CHPMe$_2$Ph and [N (SiMe$_3$)$_2$]$_3$U=CHPPh$_3$, using formally dianionic ylidic ligands[31,34]. Electronically, the U=C units found in these methanediide ([$\kappa^3$-CL$_2$]$^{2-}$) or ylidic ([C(H)(PR$_3$)]$^{2-}$) examples are best described as highly polarized, nucleophilic carbenes with σ and modest π bonding overlap between uranium and carbon. The α-carbon bonded heteroatom(s) (e.g., phosphorous) aids the delocalization of the carbon-centered charge accumulation. While compounds possessing unsupported U=C bonds are difficult to prepare under typical synthetic conditions, Andrews and co-workers have identified alkylidene and even alkylidyne species such as H$_2$C=UX$_2$ and HC≡UX$_3$ (X= H, F, Cl or Br) in low-temperature noble-gas matrix isolation experiments, providing evidence for such bonding motifs[35–37]. However, these compounds are too reactive to be isolated under typical synthetic conditions. The knowledge gained from studying stable compounds featuring U=C multiple-bonds can be extended to uranium carbide ceramics, which offer improved thermal density and higher conductivity over current UO$_2$ nuclear fuels[38–40].

Based on our recent success with the isolation of new actinide endohedral fullerenes, we turned our attention towards the synthesis and isolation of carbon cages encapsulating clusters that possess U=C bonds. The internal hollow cavity of C$_{2n}$-fullerenes can encapsulate and stabilize novel metallic clusters, especially some which are highly reactive and virtually impossible to prepare independently[41–43]. Recently, we reported the first crystallographically characterized examples of actinide endohedral metallofullerenes (Th@C$_{82}$ and U@C$_{2n}$, 2n = 74, 82) and described their unique electronic properties[43,44]. We therefore hypothesized that fullerene cages would provide an ideal architecture and electronic environment to trap and stabilize unique actinide clusters with novel bonding motifs.

Herein, using the I$_h$(7)-C$_{80}$ fullerene cage as a molecular nanocontainer, we report, to the best of our knowledge, the first structurally characterized example of unsupported uranium U=C bonds found in UCU@I$_h$(7)-C$_{80}$, possessing the unprecedented dimetallic-carbide cluster U=C=U. X-ray crystallographic analysis reveals two very short U=C bonds of 2.033(5)/2.028(5) Å, with an unexpected nonlinear U=C=U bond angle of 142.8(3)°.

## Results

**Synthesis of U$_2$C@C$_{80}$.** U$_2$C@C$_{80}$ was synthesized by the Krätschmer–Huffman arc discharge method[45]. Graphite rods, packed with graphite and finely dispersed U$_3$O$_8$, were vaporized in an arcing chamber under a He atmosphere. The compound

was isolated and purified using a multistage high-performance liquid-chromatography protocol (HPLC). The composition and purity of the isolated U$_2$C@C$_{80}$ was confirmed by high-resolution matrix-assisted laser desorption-ionization time-of-flight positive-ion-mode mass spectrometry (MALDI-TOF/MS), which presents a prominent molecular ion peak with a mass-to-charge ratio of $m/z = 1448.103$ (Supplementary Fig. 1), corresponding to the [U$_2$C$_{81}$]$^+$ empirical formula. The mass spectral isotopic distribution pattern matches the theoretically predicted one, thus confirming the molecular composition. In addition, an energy dispersive spectroscopic analysis of the purified sample was employed to determine the elementary composition of the compound. The spectrum shows characteristic peaks of uranium and carbon (Supplementary Fig. 2), which confirmed the assignment of this molecule to U$_2$C$_{81}$. Moreover, the powder X-ray diffraction pattern of U$_2$C@C$_{80}$ proves that the purified sample of U$_2$C@C$_{80}$ used for the experimental characterizations below, exists as a pure phase (Supplementary Fig. 5).

**Molecular structure of UCU@I$_h$(7)-C$_{80}$·[Ni$^{II}$-OEP].** The molecular structure of UCU@C$_{80}$ was determined by single-crystal X-ray diffraction analysis. Slow diffusion of nickel(II) octaethylporphyrin (Ni$^{II}$-OEP) in benzene into a CS$_2$ solution of UCU@C$_{80}$ yielded black UCU@I$_h$(7)-C$_{80}$·[Ni$^{II}$-OEP] cocrystals (1). In cocrystal 1, UCU@C$_{80}$ is observed to adopt a slightly distorted icosahedral I$_h$(7)-C$_{80}$ cage structure (Fig. 1a) that is highly ordered. Similarly, inside the fullerene cage, the central carbon atom C$_0$ of the endohedral UC$_0$U unit is fully ordered, while the U atoms are slightly disordered. Two major U positions (U1 and U2) have common dominant occupancy of 0.853(3), while the residual occupancies are located on both sides within ca. ½ to 1 Å distance from the two main positions, possibly indicating some large amplitude motions of the UC$_0$U cluster (Supplementary Fig. 6). The distances between the major U1,2 sites and the carbons of the adjacent aromatic rings of the cage all lie within a narrow range of around 2.50 Å (2.471(5) to 2.543(5) Å, Fig. 1b), corresponding to distances of the ring centers Cti to Ui of 2.042(1) Å and angles Cti−Ui−C$_0$≈159(1)°. The U−C$_0$ bonds are 2.03 Å (U1−C$_0$ = 2.033(5) Å, U2−C$_0$ = 2.028(5) Å), forming a nearly linear Ct1−U1—U2−Ct2 chain (Fig. 1b) bridged by a nonlinear –C$_0$– carbide anion.

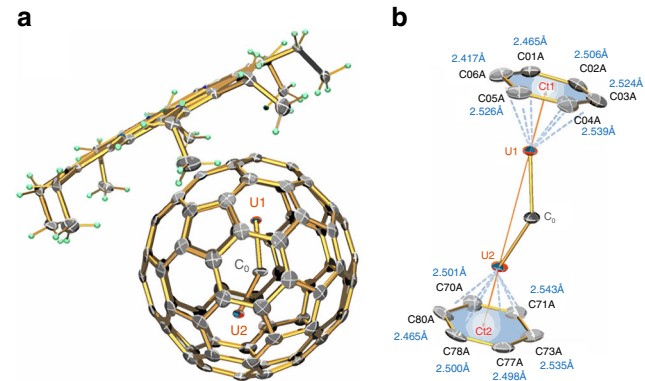

**Fig. 1** ORTEP drawing of UCU@I$_h$(7)-C$_{80}$·[Ni$^{II}$-OEP] with 40% probability ellipsoids. **a** UCU@I$_h$(7)-C$_{80}$·[Ni$^{II}$-OEP] structure showing the relationship between the fullerene cage and the [Ni$^{II}$-OEP] ligands. The two U1/U2 sites have common occupancy of 0.853(3). Four minor U sites (Supplementary Fig. 6) and the solvent molecules are omitted here for clarity. **b** Fragment view showing the interaction of the major U1-C$_0$-U2 cluster with the closest aromatic ring fragments of the cage with centers Ct1 and Ct2. The orange line connects Ct1-U1-U2-Ct2

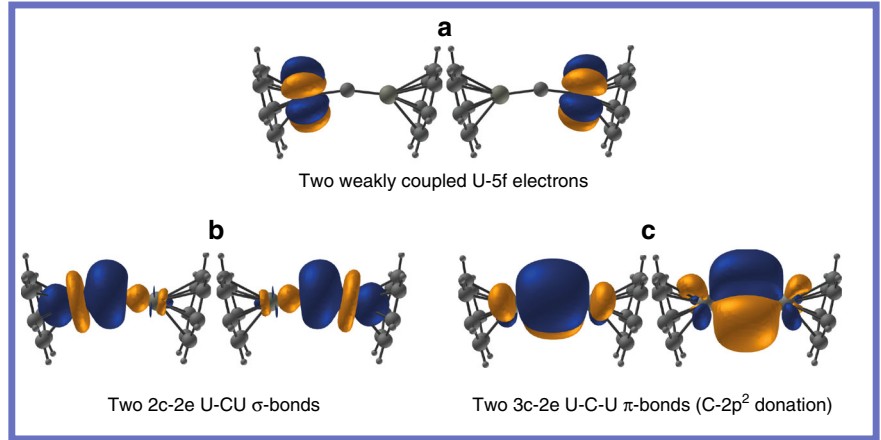

**Fig. 2** Natural Localized Molecular Orbitals (NLMOs) of sandwiched UCU. The UCU@$(C_7H_7)_2$ molecule formally consists of a $(UCU)^{6+}$ unit between two aromatic $(C_7H_7)^{3-}$ rings. Spin–orbit coupled multi-reference all-electron relativistic wavefunction calculations (state-averaged CASSCF) were analyzed with the NBO program [S27]. The contour values of the NLMOs generated thereby are $\pm 0.042\sqrt{(e/Å^3)}$. **a** The two 1-center U-5f$^1$-type NLMOs, each occupied by one electron. **b** The U–C (left) and C–U(right) 2-center 2-electron (2c2e) σ-pair NLMOs, each consisting of 67% C-2sp$^1$, 31% U-5f$^{1/2}$ 6d and 2% tail-contribution from the other U. **c** The vertical π-type and the in-plane dominantly π-type U–C–U 3-center 2-electron (3c2e) NLMOs of 65% C-2p$_s$ and 17% of each of the two U-5f$^{1/2}$ 6d

---

**Table 1 Calculated and experimentally derived geometric parameters of UCU groups**

| Geometric parameter | Calc.[a] UCU@$(3I)_2$ (molecule) | Calc.[a] UCU@$(C_7H_7)_2$ (molecule) | Calc.[a] UCU@$I_h$(7)-$C_{80}$ (molecule) | Exptl.[b] UCU@$I_h$(7)-$C_{80}$ •[Ni$^{II}$-OEP] (crystal) |
|---|---|---|---|---|
| U–C /Å | 2.024 | 2.074 | 2.022 | 2.033/2.028 |
| U•••U /Å | 3.918 | 4.118 | 3.917 | 3.849 |
| Ligand→U /Å | 2.33[c] | 2.52–2.55 | 2.47–2.51 | 2.47–2.54 |
| U–C–U /° | 150.9 | 166.1 | 151.3 | 142.8 |
| Fullerene C–C /Å | — | — | 100 × 1.43 ± 0.01 | 110 × 1.43 ± 0.03 |
| 'π-Donating' C–C /Å | — | 14 × 1.43 ± 0.01 | 20 × 1.46 ± 0.01 | 10 × 1.48 ± 0.01 |

[a]From scalar relativistic ZORA Kohn–Sham PBE VTZp approaches
[b]From crystal structure analysis by X-ray diffraction
[c]For comparison of I−U with C−U, we have subtracted the I−C bond-radii difference of 0.55 Å from the actual I−U distance

---

For hexa-coordinated uranium, Pyykkö found that two short and strong axial and four long and weak equatorial bonds are energetically near-equivalent to two long and weak axial and four short and strong equatorial bonds[46,47]. Therefore, particularly short axial bonds are expected when additional equatorial coordination is absent, as in the present case. Indeed, most U-Ct(arene) distances are reported between 2.5 and 2.9 Å, for instance 2.651(4) to 2.698(4) Å for the dinuclear U(V) inverse sandwich complex $[\{U^V(Ts)\}_2(\eta^6:\eta^6\text{-}C_6H_5Me)]$[48]. However, by far the most salient features of **1** are the unusually short U–$C_0$ bond lengths, (U1–$C_0$ = 2.033(5) Å, U2–$C_0$ = 2.028(5) Å) and the unusual bond angle of 142.8(3)° for U1–$C_0$–U2 (Table 1). In fact, the shortest U–C bond distance in poly-coordinated U-complexes to date is 2.184(3) Å, reported by Liddle et al. for the U(VI) carbene complex U=[C(Ph$_2$PNSiMe$_3$)$_2$](O)Cl$_2$[27]. Interestingly, the U–C bond lengths of the U$C_0$U unit are between the 2.067(7) Å and 1.948(7) Å, theoretically calculated distances for uranium alkylidene and alkylidyne complexes U=CH$_2$(H)F and U≡CH (Cl)$_3$[35–37], and are nearly identical to the 2.01 Å distance calculated from the sum of the covalent radii[49]. Furthermore, the U–$C_0$ distances in **1** are longer than the triple bond of the U≡O unit (1.78 Å) in the uranyl dication UO$_2^{2+}$, the U≡NR bonds (1.844(6) Å) of U(N$^t$Bu)$_2$(I)(THF)$_2$), and the terminal U≡N bond (1.83(2) Å) for the uranium nitride complex (Tren$^{TIPS}$)UN), even when accounting for differences in the atomic radii of O, N, and C of ca. 0.05 Å each[18,23,50].

Altogether, the bond lengths observed for **1** strongly support an Arene≡U=C=U≡Arene structure possessing two uranium–carbon double bonds formed by a bridging carbide atom, formally $C^{4-}$. The bending of the U=C=U cluster of **1** was an unexpected exception to the common linear geometries of $sp_\sigma^1$–$p_\pi^2$ hybridized carbon atoms found for main-group molecules of type E=C=E (E=O, NR, CR$_2$), though a few cases with angles between 180° and 90° have been theoretically proposed and observed in recent years[51].

**Computational studies of molecular structure and bonding in UCU@$I_h$(7)-$C_{80}$.** Quantum-chemical calculations were performed to further verify the unique structural parameters of UCU@$I_h$(7)-$C_{80}$ and gain insight about the bonding of the encapsulated bent U=C=U cluster. The molecular structure was modeled for free UCU@$I_h$(7)-$C_{80}$ and for LUCUL (L = C$_7$H$_7$, 3I) model molecules, using quasi-relativistic density-functional (DF) approximations (Supplementary Figs. 13, 14) and ab-initio CASSCF(10e,12o)-PT2 approaches (Table 1). Additional information of the structure, orbitals and electronic states of LUCUL is available in Supplementary Figs. 15–19 and Supplementary Tables 3–5. For the three calculated cases, the formal oxidation state[52] is identified as 5+ for each U, derived from the formal [L$^{3-}$≡>U$^{5+}$<=>C$^{4-}$=>U$^{5+}$<≡L$^{3-}$] Lewis structure, where L$^{3-}$≡>stands for the cage arene units that accept 3 electrons from

each uranium atom and forms 3 dative pair bonds with the nearly empty U-5f6d valence shell. In the case of UCU@$I_h$(7)-$C_{80}$, the two $L^{3-}$ units are the ligating arene units of the encapsulating $C_{80}^{6-}$ closed-shell cage. The C and I atoms of the model units ($C_7H_7$) and (3I) have similar electronegativity as C of ($C_{80}$), and the model units also become closed-shell ligands upon accepting 3 electrons from each U. The effective physical partial charges in all three cases remain small, around $-1$ for formal $C^{4-}$, around $+\frac{1}{2}$ to $+1$ for formal $U^{5+}$, and small for the ligand atoms of $C_{80}$, $2C_7H_7$ or 6I (Supplementary Table 1).

The atomic free valence and the spin and orbital populations (Supplementary Table 1) indicate a single f-electron on each U atom. The experimental U–$C_0$ bond lengths, though unprecedentedly short (2.03 Å), are well reproduced in all three computed models (2.02 Å for ligands $C_{80}$ or ($I_3$)$_2$, 2.07 Å for ($C_7H_7$)$_2$). Further, the observed unusual U–$C_0$–U bond angle of 142.8° is comparable to those computed for UCU@$C_{80}$ (151.3°) and (3I) UCU(3I) (150.9°), while the bulkier ligands of ($C_7H_7$)UCU($C_7H_7$) led to 166.1°, which is still far from linear (Table 1). Neither the bond lengths nor the bond angle of the UCU unit appear to be significantly distorted by the encapsulation in $C_{80}$.

We chose the localized equivalent molecular orbital picture of the two simpler model systems to sketch the correlated and spin–orbit coupled valence electronic structure of ligated $UC_0U$. ($C_7H_7$)UCU($C_7H_7$) (Fig. 2) and (3I)UCU(3I) (Supplementary Fig. 15) gave very similar results. Each U forms a strong σ-bond to the central $C_0$ atom through covalent overlap of hybridized U (5 f6d) orbitals with C(2s2p) hybrids. The U–C bonds of dominant π-character can be described as three-center two-electron (3c2e) bonds that possess some σ-admixture. The large electronegativity difference of C and U (1.3 Pauling units) leads to bond-pair polarization toward $C_0$, especially for the π-type pairs, which are strongly localized on $C_0$ and lead to its effective negative charge. The U–C interactions are characterized by Mayer bond orders (BO) of 1.4 from DF calculations (comparable to 1.6 for the U=As double bond in [K(B15C5)$_2$] {[N (CH$_2$CH$_2$NSi$^i$Pr$_3$)$_3$] U=AsH})[26] and by a Roos effective bond order (EBO) of 1.9 (counting the formally bonding—antibonding NOs from our CASSCF calculation of ($C_7H_7$)UCU($C_7H_7$), which substantiates the U=C double bond character).

The bonding motif is nicely illustrated by the electron localization function (ELF) map in Fig. 3, which shows the electron–density accumulation of the two two-center U–C σ-bonds and of the in-plane three-center U–C–U π-type bond, as well as the multiple maxima of the inner-valence non-bonding 5f electrons. Thus, the bending of the U=C=U unit can be interpreted in terms of the localized negative charge build-up on $C_0$, which generates lone electron-pair density and gives rise to an sp$^1$/sp$^2$-hybridized type geometry at carbon with bond angles in the middle between 120° and 180°. This charge accumulation is partially offset by donation into the U(5f6d) orbitals. Thus the

bonding model can be described as highly polarized U=C interactions with partial π-overlap strengthened by electrostatic attractive forces between the cationic $U^{5+}$ atoms and the anionic $C_0^{4-}$ carbide bridge.

**Spectroscopic properties of UCU@$I_h$(7)-$C_{80}$.** Nuclear Magnetic Resonance (NMR) spectroscopy was used to characterize UCU@$I_h$(7)-$C_{80}$. The $^{13}$C NMR spectrum of UCU@$I_h$(7)–$C_{80}$ measured at 298 K shows only two sharp signals at 138.53 and 125.14 ppm with a 3:1 intensity ratio, corresponding to the sets of 60 and 20 equivalent carbon atoms of the unperturbed $I_h$–$C_{80}$ fullerene cage (Supplementary Fig. 7). The apparent symmetry is likely due to fast and large-amplitude librations of the cage, similar to those proposed for $M_2$@$I_h$-$C_{80}$ (M = La, Ce) and $M_3N$@$C_{80}$ (M = Sc, Y)[42,53,54]. Unfortunately, no signal was detected for the single bridging $C_0$ carbon atom, likely due to limited sample amount, as well as to paramagnetic broadening effects from the unpaired 5f$^1$ electrons on each uranium. Noteworthy is that the $^{13}$C NMR spectrum measured at lower temperature (i.e., 283 K) shows slightly shifted signals at 138.35 and 124.65 ppm, respectively, with a slightly larger chemical shift difference ($\Delta\delta = 13.70$ ppm) than those observed at 298 K ($\Delta\delta = 13.39$ ppm). Such a trend is very similar to that observed for $Ce_2$@$I_h$-$C_{80}$[55], consistent with the paramagnetic nature of UCU (see below) and with the presence of an unpaired f electron on the formal $U^{5+}$ (5f$^1$6d$^0$7s$^0$) ions (Supplementary Table 1)[54]. On the other hand, the slight temperature dependence of $\Delta\delta/\Delta T = 0.02$ ppm/K might also be accounted for by the libration of the UCU cluster that neutralizes the paramagnetic effects (mainly pseudocontact interactions) for all carbon atoms.

We also utilized Fourier transform infrared absorption (FTIR), Raman emission and photoluminescence (PL) spectroscopies together with quantum-chemical calculations to further characterize UCU@$I_h$(7)-$C_{80}$. An experimental and theoretical spectral overview is presented in Fig. 4. The high-wavenumber range from 1600 to 1100 cm$^{-1}$ contains 104 carbon cage C–C stretching vibrations, resembling those of other $I_h$-$C_{80}$ based endohedral fullerenes such as $Ln_3N$@$C_{80}$[41,56]. Typical characteristics are the major overlaid bands around 1380 cm$^{-1}$ and the featureless gap between 1100 and 900 cm$^{-1}$. The remaining 130 collective deformations originating from bending and torsional motions of the carbon cage atoms show up between 900 and 200 cm$^{-1}$.

Quantum-chemical calculations suggest that an endohedral cluster of $n$ atoms (here $n = 3$) shows 3 frustrated translational rocking modes against the cage (at 33, 59, and 115 cm$^{-1}$), 3 frustrated torsional wagging modes (at 16, 50, and 142 cm$^{-1}$), and $3n – 6 = 3$ internal vibrations. The latter comprise the $UC_0U$ bending mode (at 97 cm$^{-1}$), the symmetric stretch (coupled to near-degenerate cage deformations from 250 to 300 cm$^{-1}$), and the asymmetric stretch (at 780 cm$^{-1}$). The observed weak features

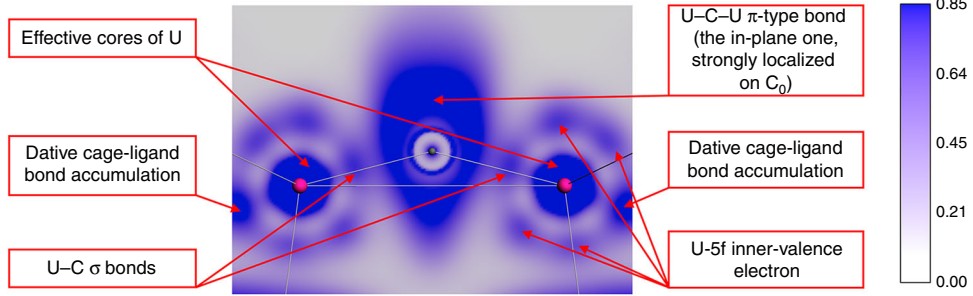

**Fig. 3** ELF-plot of ligated UCU in the molecular plane. The left and right ligands are two ($I_3$) groups. ELF is the 'electron localization function' of Becke and Edgecombe, showing electronic accumulations in the inner atomic shells, in the atomic lone-pair regions, and in covalent and dative bond regions

in the Raman spectrum above 100 cm$^{-1}$ at 126, 148, and 277 cm$^{-1}$ agree reasonably well with the predicted UCU modes given above (in bold italics, estimated as weakly Raman active). Notably, the asymmetric UCU stretch appears as a pronounced feature in the IR spectrum at 785 cm$^{-1}$, which is consistent with the high negative charge on the $C_0$ atom of the endohedral UCU cluster, and with the crystallographic observation of very short axial U=C bonds.

Further, an experimental violet-blue photo-luminescence progression of $U_2C@I_h(7)$-$C_{80}$ starting at a wavelength of 430 nm (~2.95 eV) in steps of ca. 1425 cm$^{-1}$ (Supplementary Fig. 8) may originate from a ligand-to-metal charge-transfer excitation of the highest occupied molecular orbital, which has dominant density on the fullerene cage near the U atoms. There are also $C_{ring}$-breathing vibrations around 1400 cm$^{-1}$ localized in the spatial $C_{ring}$–U–C–U–$C_{ring}$ region.

The redox properties of UCU@$I_h(7)$-$C_{80}$, as investigated by means of cyclic voltammetry, show a surprisingly small electrochemical gap of only 0.83 eV (Table 2, Supplementary Fig. 9). Most reported transition metal cluster fullerenes of $I_h(7)$-$C_{80}$ have gaps larger than 1 eV[41]. Typical M$_3$N@$I_h(7)$-$C_{80}$ fullerenes have gaps of 1.8–2.2 V. In particular, the first reduction potential of UCU@$I_h(7)$-$C_{80}$ (−0.41 V) is much more positive than that of

any other reported M$_3$N@$I_h(7)$-$C_{80}$ fullerene[57], e.g., −1.26 V for Sc$_3$N@$I_h(7)$-$C_{80}$[58] and −0.94 V for TiLu$_2$C@$I_h(7)$-$C_{80}$[59]. UCU@$I_h(7)$-$C_{80}$ obviously exhibits a much better electron-accepting ability than other $I_h(7)$-$C_{80}$ clusterfullerenes. This significant difference between UCU@$I_h(7)$-$C_{80}$ and other reported M$_3$N@$C_{80}$ fullerenes indicates a major difference of their electronic structures, which may be caused by the encapsulation of the U=C0=U cluster with an open d–f shell electronically interacting with the cage. Interestingly, when compared to most other $I_h(7)$-$C_{80}$ cluster fullerenes, the redox potentials of UCU@$I_h(7)$-$C_{80}$ are close to those of dilanthanide-fullerene, such as La$_2$@$I_h(7)$-$C_{80}$, which exhibits an electrochemical gap of 0.87 eV[60]. In general, the electrochemical gaps are related to the HOMO-LUMO gaps of the closed-shell fullerene subsystems. However, in the present case, the encapsulated U=$C_0$=U cluster has a low-lying open U-5f shell, causing an abnormally small f–f gap of the UCU@$I_h(7)$-$C_{80}$ molecule. The reversible reduction of UCU@$I_h(7)$-$C_{80}$ is also quite remarkable, since M$_3$N@$I_h(7)$-$C_{80}$ fullerenes with group 3 metals exhibit irreversible reductions[57].

The XPS spectrum of UCU@$I_h(7)$-$C_{80}$ was also recorded (Supplementary Fig. 10). The U-4f$_{7/2}$ ionization appears at 378.6 eV. In general, ionization energies increase with the formal oxidation state, e.g., from the U-4f$_{7/2}$ value of ca. 377$_{1/3}$ eV for the neutral uranium metal to ca. 381$_{3/4}$ eV for some uranyl(VI) salts. The present value of 378.6 eV is consistent with the very low effective charges on U as computed for UC$_0$U@$C_{80}$ (see below, U ~ +1.0e; C$_o$ ~ −1.0e; C$_{fullerene}$ ~ ± 0.0e), caused by the strong electron pair donation of formal C$_{80}^{6-}$ and C$_0^{4-}$ to the formal U$^{5+}$ ions.

To gain further insight into the unique electronic characteristics of the U=C=U cluster in UCU@$C_{80}$, the magnetic susceptibility was measured (see Supplementary Fig.11 and Methods). The magnetic curves exhibit two regimes, (i) one at "high temperatures" from 300 K down to ca. 60 K, and (ii) another one at "low temperatures" from ca. 40 K down to 2 K. The "high temperature" region exhibits a huge, basically temperature-independent paramagnetism (TIP) of yet unknown origin. It is unknown for uranium complexes with common ligands[21,22,48,61–65] and may possibly be due to the cooperative coupling of UCU and $C_{80}$ in the solid and related to the low band-gap calculated for UCU@$C_{80}$ (see Supplementary "Quantum Computational Methods").

At "low temperatures", we observed a Curie type paramagnetism ~$\mu^2$/T on top of the TIP, with net $\mu$-values increasing with temperature from below to above a $\mu_{Bohr}$, which appears compatible with hardly coupled unique units of type $\eta^3$L≡U (V)=C=U(V)≡L$\eta^3$, i.e., the endohedral di-uranium carbide fullerene molecules. The "low-temperature" literature values of $\mu_{eff}$[21,22,48,61–65] for mono- and bridged di-metallic U(V) and U (IV) complexes, typically increase, too, from low values up to 3 $\mu_{Bohr}$. It shows the expected temperature dependent weak coupling of two spins and with the two partially quenched U-f±3 orbital angular momenta, the experimental curves being

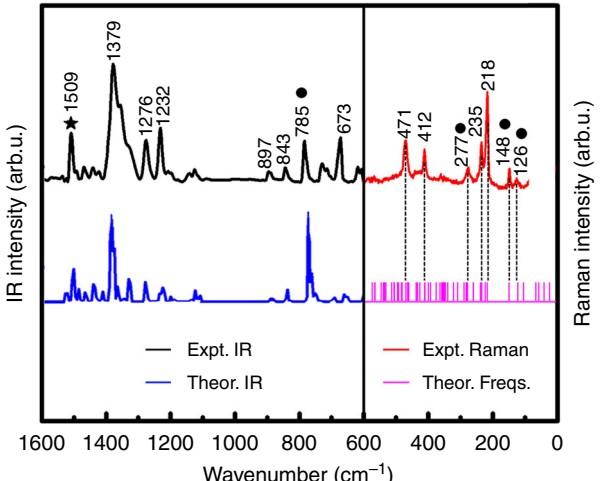

**Fig. 4** Observed and theoretically predicted vibrational features of UCU@$I_h(7)$-$C_{80}$. The left upper curve (black) presents the observed Infrared absorption (IR) spectrum vs. wavenumber from 1600 to 600 cm$^{-1}$, with quantum-chemical density-functional simulation below (in blue). On the right is the observed Raman emission (in red) vs. wavenumber from 600 to 100 cm$^{-1}$, with calculated wavenumbers ('Theor. Freqs.') in the 600–0 cm$^{-1}$ range (below in magenta). Observed local UCU vibrations above 100 cm$^{-1}$ are indicated by heavy dots. Observed and calculated spectral C$_{80}$-cage features represent either a single mode, or an overlay of several near-degenerate ones. indicates a contamination by CS$_2$

| Table 2 Redox potentials and electrochemical gaps of UCU@$I_h(7)$-$C_{80}$, La$_2$@$I_h(7)$-$C_{80}$, and Sc$_3$N@$I_h(7)$-$C_{80}$ | | | | | | | |
|---|---|---|---|---|---|---|---|
| **Fullerenes** | $E^{2+/+}$ a | $E^{+/0}$ a | $E^{0/-}$ a | $E^{-/2-}$ a | $E^{2-/3-}$ a | $E_{gap}$ b | Ref. |
| UCU@$I_h(7)$-$C_{80}$ | +1.05$^d$ | +0.42$^c$ | −0.41$^c$ | −1.34$^c$ | | 0.83 | This work |
| La$_2$@$I_h(7)$-$C_{80}$ | +0.95$^c$ | +0.56$^c$ | −0.31$^c$ | −1.72$^c$ | −2.13$^d$ | 0.87 | 60 |
| Sc$_3$N@$I_h(7)$-$C_{80}$ | | +0.97$^c$ | −1.26$^d$ | −1.62$^d$ | −1.82$^d$ | 2.23 | 58 |

$^a$Redox potentials in V vs. ferrocene couple
$^b$Electrochemical gaps in eV
$^c$Half-wave potential (reversible redox process)
$^d$Peak potential (irreversible redox process)

reasonably reproduced by numerical simulations (see Supplementary Method "Quasi-relativistic correlated ab initio approaches" on the $(C_7H_7)UCU(C_7H_7)$ model).

Attempts to further resolve the electronic structure using EPR spectroscopy were unsuccessful, as no clearly defined signal was observed at 4 K (Supplementary Fig. 12). Factors such as near degenerate electronic states and cage shielding effects as well as the huge TIP complicate the analysis. Thus, additional studies of the electronic properties of this unique and unprecedented system are warranted and will be communicated in due time.

## Discussion

In summary, the overall agreement between the crystallographic, luminescence, Raman, IR, core-electronic, magnetic and voltammetric results, and the quantum-computational findings, conclusively show that a pure $UCU@I_h(7)$-$C_{80}$ compound was synthesized, which consists of a weakly perturbed $I_h$-$C_{80}$ cage and a sandwiched, bent and strongly bound polar endohedral near-symmetric U=C=U unit. The quantum-chemical results suggest that both U atoms in the UCU unit have a formal +5 oxidation state. The encapsulation causes little bond length or angle deformation of the UCU fragment. The encapsulation protects the axially ligated U atoms from binding with harder or more electronegative ligands like O or N (as compared to carbon in the forms of $C_0^{4-}$ and $C_{80}^{6-}$) so that $C_0$ can participate in primarily axial $L\equiv>U=C_0$ bonding at unusually short $U–C_0$ distances of 2.03 Å, and is not pushed into the weaker bonding equatorial plane of an E=U=E building block. The discovery of unsupported U=C bonding in a molecular compound confirms the distinction between "diagonal" and "poly-coordinated" uranium, and between "axial" covalent and "equatorial" dative bonding mechanisms. The work reported here offers a deeper understanding of the fundamentals of uranium bonding properties. This study also demonstrates that fullerene cages can be utilized as effective nanocontainers to stabilize and study rare and reactive clusters which contain actinide metal–ligand bonds.

## Methods

**Synthesis, separation and purification of $U_2C@I_h(7)$-$C_{80}$.** A soot containing uranium fullerenes was synthesized by a direct-current arc discharge method. Graphite rods of $U_3O_8$ and graphite powder were annealed in a tube furnace at 1000 °C for 20 h under an Ar atmosphere and then burned in the arcing chamber under a 300 Torr He atmosphere. In total 1.87 g of graphite powder and 1.83 g of $U_3O_8$ (24:1 atom number ratio) were packed in each rod (6.7 g, without filling). The collected raw soot was refluxed in chlorobenzene under Ar atmosphere for 12 h and the solution was filtered. After solvent removal, the fullerenes were extracted by dissolving in toluene. On average ca. 140 mg of crude fullerene mixture per rod was obtained. In total, 60 carbon rods were burned in this work. Separation and purification of $U_2C@I_h(7)$-$C_{80}$ was achieved by a three-stage HPLC procedure (Supplementary Fig. 3), using columns including a Buckyprep M column ($25 \times 250$ mm$^2$, Cosmosil, Nacalai Tesque, Japan), a Buckyprep D column ($10 \times 250$ mm$^2$, Cosmosil, Nacalai Tesque, Japan) and a Buckyprep column ($10 \times 250$ mm$^2$, Cosmosil, Nacalai Tesque, Japan). Toluene was used as the mobile phase and the UV detector was adjusted to 310 nm for fullerene detection. The HPLC procedures and corresponding MALDI-TOF spectra for the isolated fractions are shown in Supplementary Figs. 3, 4. In total, ca. 2 mg of highly purified $U_2C@C_{80}$ was obtained for characterization.

**Single crystal and powder X-ray diffraction analysis.** Black co-crystals of $UCU@I_h$-$C_{80}\cdot[Ni^{II}$-OEP]) were obtained by allowing the benzene solution of [$Ni^{II}$-OEP](1.2 mg/mL, 0.8 mL) and the $CS_2$ solution of $UCU@I_h$-$C_{80}$(1 mg/mL, 0.5 mL) to diffuse together. X-ray diffraction data were collected at 120 K using a diffractometer (APEX II; Bruker Analytik GmbH) equipped with a CCD collector. The Multiscan method was used for the absorption correction. The structure was resolved using direct methods (SIR2004) and refined on $F^2$ with the full-matrix least-squares approach using SHELXL2014[66,67] within the WinGX package[68].

Crystal Data for $UCU@I_h$-$C_{80}\cdot[Ni^{II}OEP]\cdot1.5C_6H_6\cdot CS_2$: $C_{127}H_{53}N_4NiS_2U_2$, $M =$ 2159.70, $0.05 \times 0.05 \times 0.03$ mm$^3$, monoclinic, P $21/c$ (No. 14), $a = 17.678(4)$ Å, $b =$ 16.970(3) Å, $c = 26.695(5)$ Å, $\beta = 106.65(3)^\circ$, $V = 7673(3)$ Å$^3$, $Z = 4$, $\rho_{calcd} =$ 1.934 g cm$^{-3}$, $\mu = 4.578$ mm$^{-1}$, $\theta = 2.688–27.092^\circ$, $T = 120$ K, $R_1 = 0.0610$, $wR_2 =$

0.1045 for all data; $R_1 = 0.0388$, $wR_2 = 0.0610$ for 16,771 reflections ($I > 2.0\sigma(I)$) with 1278 parameters and 784 restraints. Goodness of fit indicator 1.007. Maximum residual electron density 2.096 e Å$^{-3}$. For further details see the Supplementary Data 1, 2.

In addition, a powder X-ray diffraction analysis of $UCU@C_{80}$ was conducted at the Shanghai Synchrotron Radiation Facility (SSRF) with wavelength $\lambda = 0.6199$ Å, and the intensity vs. $2\theta$ pattern was recorded, see Supplementary Fig. 5.

**Computational methods.** Open-shell Kohn–Sham single-reference calculations were carried out with the ADF-2016, Gaussian-09 and ORCA-4.0 software, applying scalar and spin–orbit-coupled quasi-relativistic approaches (RECP or ZORA) with triple-$\zeta$ double-polarized basis sets. Bonding analyses were then performed using the NBO5.0 and Multiwfn codes. Multi-reference CASSCF, CASPT2, and RAS-SI/SO wave-function calculations were performed using the MOLCAS-V8.1 software applying the quasi-relativistic DKH Hamiltonian. Further details are given in the Supplementary "Quantum Computational Methods".

**Data availability.** The coordinates for the X-ray structure of **1** are available free of charge from the Cambridge Crystallographic Data Centre under deposition no. CCDC 1572891. All other data supporting the findings of this study are available from the corresponding authors on request.

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

## Acknowledgements

We cordially thank Prof. Xing Lu (Huazhong University of Science and Technology, China) for kind help with the crystallographic measurements and Prof. Marilyn Olmstead and Prof. Alan Balch (University of California at Davis, USA) for help with the data analysis. We also gratefully acknowledge the help with magnetic measurement by Prof. Taishan Wang and Mingzhe Nie (Institute of Chemistry, The Chinese Academy of Sciences) and helpful discussions of the magnetism by Prof. Stephen Liddle (University of Manchester, England), Prof. Thomas Albrecht-Schmitt and Dr. Ryan Baumbach (Florida State University, Tallahassee, USA) and Prof. Jörn Schmedt auf der Günne (University of Siegen, Germany). We thank Prof. Lin Wang and Dr. Cuiying Pei (Center for High Pressure Science &Technology Advanced Research) for the help with Powder XRD measurements. N.C. and L.F. thank the support from the NSFC (51772196, 51772195), the NSF of Jiangsu Province (BK20171211), Priority Academic Program Development of Jiangsu Higher Education Institutions (PAPD) and the project of scientific and technologic infrastructure of Suzhou (SZS201708). J.L. thanks the support from the NSFC (21433005, 91426302, 21521091, and 21590792), the Supercomputer Center of the Computer Network Information Center, the Chinese Academy of Sciences, and the Computational Chemistry Laboratory of the Department of Chemistry, Tsinghua University. L.E. thanks the US National Science Foundation (NSF) for generous support of this work under the NSF-PREM program (DMR 1205302) and the CHE-1408865. The Robert A. Welch Foundation is also gratefully acknowledged for an endowed chair to L.E. (Grant AH-0033). S.F. thanks the NSF-PREM program (DMR 1205302) and the Welch Foundation for support (AH-1922-20170325). J.A. thanks the U.S. Department of Energy, Office of Basic Energy Sciences, Heavy Element Chemistry program, grant DE-

SC0001136 for support. G.V. and A.A.P. acknowledge the European Research Council under the European Union's Horizon 2020 programme (Grant No. 648295 "GraM3") for financial support; they also thank D. S. Krylov and A. U. B. Wolter (IFW Dresden) for their help with the magnetic measurements. W.H.E.S. thanks the support from the Theoretical Chemistry Center of Tsinghua University.

## Author contributions

N.C., L.E., and X.Z. conceived and designed the experiments. X.Z. and Y.W. synthesized and isolated all the compounds. W.L., J.L., W.H.E.S., X.C., A.H., S.G., D.C.S., T.J.D., and J.A. performed the computations and theoretical analyses. L.F. performed the crystallographic analysis. N.A. and S.D. provided the XPS analysis. X.L. carried out the NMR test. G.V. and A.A.P. measured and analyzed the magnetic data. N.C., L.E., W.H.E.S., J.L., L.F., S.F., X.Z, W.L., S.G., J.A., and S.W. co-wrote the manuscript.

## Additional information

**Competing interests:** The authors declare no competing interests.

