## [Peer Review File · Nature Communications]

Reviewers' comments:

Reviewer #1 (Remarks to the Author):

The authors describe the synthesis and molecular structure of a diuranium carbide cluster encaged in a C₈₀ fullerene. The preparation of the title complex was accomplished by reaction of a graphite rod of U₃O₈ and graphite powder in a furnace at 1000°C under He atmosphere. Product purification was accomplished by HPLC and the product characterized by MALDI-TOF and X-ray diffraction. The paper is clearly written, and the experimental work is also complemented by computational studies. Overall, I believe that this contribution could appeal to the readership of Nature Communications, since actinide element multiple bonds is a field of current interest, especially the resulting insights into bonding, reactivity and structure. In contrast to d-transition metals the actinide element multiple bonds are less covalent and significantly polarized, which makes them reactive and generally difficult to isolate. The approach employed by the authors is in this respect rather innovative and tames the high reactivity of the diuranium carbide species. So while I am looking very favorable on this contribution, I also believe that additional experiments are required before this paper is finally acceptable:

1. I am missing additional information regarding the electronic structure of the diuranium carbide species. Computational studies reveal that the two U(V) centers are only weakly coupled, so with two 5f¹ systems magnetic susceptibility studies are requested to elucidate the magnetic coupling between the U(V) spin carriers.
2. What is the yield of the title complex?
3. The authors report an ¹³C NMR spectrum, which they assign to the carbon atoms of the (Ih-C₈₀)⁶⁻ moiety, while the bridging carbon atom was not observed. It would be nice to provide some comparisons from the literature support this assignment. Is there any temperature dependence in the chemical shifts, since the cluster contains two paramagnetic U(V) atoms. There should be some temperature dependence.
4. In the SI the authors report on the electrochemistry of the title compound (Fig. S11) and compare these observations to other complexes (Table S3). However, the authors do not discuss anything of these or even attempt to do so. Some comments are actually necessary to put this into perspective.

Reviewer #2 (Remarks to the Author):

This manuscript reports some rather important results and indeed deserves publication in NatComm, but it comes to me the manuscript was prepared in a hurry manner and the writing was not equal. Substantial revisions are necessary before accept.

Theoretically:

1. Although the LUCUL unit can simulate the encapsulation as discussed, the authors should give reason why they choose (C₇H₇)³⁻ and (I₃)³⁻?
2. The optimized geometrical structures of UCU@Ih(7)-C₈₀ and LUCUL should be given in the manuscript.
3. The optimizations of the three complexes in Table 1 were carried out using CASSCF (10e, 12o) method? If yes, the authors should definitely point it out in the manuscript.
4. How to confirm the formal oxidation state as +5?
5. Fig. 3, the color of ELF seems to be ambiguous, for example, the blue color in different locations.
6. Page 7, the U-C Mayer bond orders seem to be much lower compared to the formal U-C double bonds.

7. All the Figures and Tables in Supplementary Materials should be discussed in the manuscript. Experimentally:

1. The synthesis of complex 1 is quite complicated, so I would like to know how much it yields.
2. The molecular composition and structure of complex 1 were determined by MALDI-TOF/MS and single-crystal X-ray diffraction analysis. These characterization methods are not comprehensive. Elemental analysis should be included in the manuscript.
3. The purity of the isolated U₂C@C₈₀ was confirmed by MALDI-TOF/MS. However, the purity of complex 1 was not mentioned in the manuscript. The purity of the sample is the prerequisite for all characterization. Therefore, the PXRD patterns of complex 1 should be included to determine purity.
4. No signal was detected for the single bridging C₀ carbon atom in the ¹³C NMR spectrum of complex 1. The author should offer an explanation.
5. There are 11 figures in the SI, only 5 of them were mentioned in the manuscript. Is it means that the other 6 are not necessary? The author should delete these figures or add some words to describe them in the manuscript

Minor suggestions:

- (1) The label "b" is lost in Table 1.
- (2) Most of the Supplementary Materials is not referred (Fig S6-S11, Table S3) in the manuscript and some is wrong number, such as "The calculated atomic partial charges of order ± 1 are listed in Supplementary Table S3" should be ".....Table S2"

Reviewer #3 (Remarks to the Author):

This paper describes the realisation of an important synthetic result – the isolation of a molecular compound that contains unsupported U=C bonds. The compound consists of a diuranium carbide unit, U=C=U, encapsulated within a C₈₀ fullerene cage. Uranium-element multiple bonding is of immense topical interest as studies of these compounds can shed light on actinide covalency, which is of importance in actinide/lanthanide separations in nuclear fuel cycles. Furthermore, compounds containing U=E bonds can exhibit interesting reactivity profiles and are interesting for fundamental reasons.

The diuranium carbide compound in this paper is the first molecular example of such a motif, hence the journal requirements of originality, significance and impact have been met. This work would therefore be a welcome addition to the literature and it is an appropriately important result to be considered for publication in this journal. However there are some points for the authors to address before I can recommend publication:

1. Title: shouldn't contain the word "unprecedented" or "first"; this is a standard requirement as all novel results are unprecedented.

2. Introduction: the referencing is not wholly appropriate. Many references appear to be given to a somewhat random selection of primary research papers giving homage to leading names in the field, rather than a more methodical selection of appropriate reviews. For example, refs 5-7 on actinide-ligand multiple bonding lagging behind the transition metals contains one example of molecular U=P multiple bonding and two examples of matrix-isolated U=CH₂. Whilst these three references should be cited at some point in the paper the correct references at this point are surely ref #8 and Hayton's other U=E review (Dalton Trans., 2010, 39, 1145) should also be added here. Then, when stating that unsupported U=C bonds are conspicuously missing the most appropriate review by Liddle is currently missing (Comments Inorg. Chem., 2015, 35:5, 262; Liddle has written three other reviews on uranium carbenes that could also be cited, plus his comprehensive review on molecular uranium chemistry is also appropriate: Angew. Chem. Int.

Ed., 2016, 54, 2). Quite why Mazzanti's excellent work on uranium nitride functionalisation is referenced at this point (#12) is unclear. The authors should state what they mean by "unsupported", as the Hayton and Gilje ylidic examples in the next sentence are not pincer complexes, they are merely phosphorus-stabilised.

Other appropriate omissions from the introduction that require inclusion include: U=Se, =Te and =As bonds have also been isolated, with appropriate Hayton/Liddle references (bottom of page 2); a clear statement that U=C bonds have been observed by matrix isolation (top of page 3); the first uranyl(VI) carbene (Ephritikhine, JACS, 2011, 133, 6162), the first uranium(V) carbene (Liddle, Angew. Chem. Int. Ed., 2011, 50, 2383) and a reference to Gilje for the Cp₃U=C system (JACS, 1981, 103, 3589) (all top of page 3); a reference should be added to uranium carbide ceramics (middle of page 3). Finally, as this is a molecular version of uranium carbide, which has been known for over a hundred years and is of relevance to nuclear cycles, the authors should make some reference to this work for context. I am unfamiliar with this area but the review by Frad (Adv. Inorg. Chem. Radiochem., 1968, 11, 153) appears to be a good place to start.

In the last sentence of the introduction change "first example" to "first structurally characterised example" (to acknowledge matrix isolation studies). Esds should also be added to the bond lengths and angles in this last paragraph of the intro and paragraphs 1 and 2 of page 5.

3. Molecular structure discussion: The U-arene distances are compared to those in an example by Diaconescu that contains a formally dianionic C₇H₈ unit and U(III) centres. As the authors claim U(V) centres they could also compare these distances to the sole example by Liddle (Angew. Chem. Int. Ed., 2011, 50, 10388) or terminal U-arene complexes rather than inverse sandwiches (e.g. Inorg. Chim. Acta., 1971, 5:3, 439 and Cotton examples in the 1980s). The Liddle reference in paragraph 2 on page 5 should be #13, not #14. Finally, Boncella's RN=U=NR uranyl analogue complex (Science, 2005, 310, 1941) should be referenced at the bottom of page 5 as these bond lengths (1.844(6) Å) are also shorter than the U=C bonds herein.

4. Spectroscopic properties: the chemical shift range for the ¹³C NMR experiment should be provided; U(V) complexes can give unusual NMR shifts so it is important for the authors to state this. Whilst I am unsurprised that no signal was observed it could be as deshielded as 500 ppm. Furthermore, as the authors have concluded U(V) centres are present they should perform standard characterisation techniques to confirm this and show the highest levels of rigour, which can include: (i) NIR spectroscopy. The spectrum currently provided in the SI has not been converted to give extinction coefficients; could the authors provide concentrations as concentrated U(V) solutions often give absorbances at ca. 6700 cm⁻¹ (~1250 nm) for an f-f transition; (ii) VT DC measurements from 300 K - 2 K using a SQUID can confirm the magnetic 5f¹ U(V) ground state as even at 2 K there will be a significant magnetic susceptibility; (iii) EPR spectroscopy can also confirm the 5f¹ configuration; 5-10 K should be the ideal temperature to carry these experiments out.

5. Conclusions: the authors state that they have discovered "genuine" U=C bonding. I think that as written this sounds a little disparaging of previous work (why are the matrix isolation experiments not genuine, or the phosphorus-stabilised work?), so should be rephrased as this work is complementary. Furthermore, this is not an "elusive" uranium-element multiple bond, as the authors have currently stated.

6. Methods: Although I expect that the yields of this product are extremely low this has not currently been stated and there is currently not enough detail for another group to repeat this work. Therefore, the authors should add how much U₃O₈ and graphite powder was used in grams, together with the crude yield in g, and the crystalline yield in g and mmol. I imagine that there is a sub-1% yield based on U₃O₈, so no need for percentage yields if so.

According to the reviewer's comments, we made the following changes in the manuscript.

Reviewer #1 (Remarks to the Author):

The authors describe the synthesis and molecular structure of a diuranium carbide cluster encaged in a C₈₀ fullerene. The preparation of the title complex was accomplished by reaction of a graphite rode of U₃O₈ and graphite powder in a furnace at 1000°C under He atmosphere. Product purification was accomplished by HPLC and the product characterized by MALDI-TOF and X-ray diffraction. The paper is clearly written, and the experimental work is also complemented by computational studies. Overall, I believe that this contribution could appeal to the readership of Nature Communications, since actinide element multiple bonds is a field of current interest, especially the resulting insights into bonding, reactivity and structure. In contrast to d-transition metals the actinide element multiple bonds are less covalent and significantly polarized, which makes them reactive and generally difficult to isolate. The approach employed by the authors is in this respect rather innovative and tames the high reactivity of the diuranium carbide species. So while I am looking very favorable on this contribution, I also believe that additional experiments are required before this paper is finally acceptable:

1. I am missing additional information regarding the electronic structure of the diuranium carbide species. Computational studies reveal that the two U(V) centers are only weakly coupled, so with two 5f¹ systems magnetic susceptibility studies are requested to elucidate the magnetic coupling between the U(V) spin carriers.

Reply: Special thanks to the reviewer for this suggestion. We have performed magnetic measurements and obtained amazing very interesting results. SQUID measurements between 2 and 300 K revealed an unprecedented temperature-independent paramagnetism (TIP) that will be elucidated through further experiments in the future. It is clearly visible from 75 to 300 K. On top of that we found an additional, more common Curie-type paramagnetism, clearly separable between 2 and 25 K. Due to the large TIP background, however, the effective moment can only be roughly estimated as increasing from below 1 μ_B at 2 K to above 1.5 μ_B at higher T. Such behavior is found for various U(V) as well as U(IV)-U(VI) complexes. We have also investigated the EPR features at 4 K, but these were extremely broad from 2 to 7 kG without any sharp lines. We have inserted our findings on p. 11 of the revised MS. See also the SI, pp. S3, S11 & S12.

2. What is the yield of the title complex?

Reply: We didn't give the exact yield of the title complex ($\text{UCU}@C_{80}$) because the synthetic method is a 'physical procedure' rather than a typical chemical synthetic process, and the yield is difficult to estimate and much lower than 1%. But to address the reviewer's concern, we added more experimental details in the section of 'Synthesis, separation and purification of $\text{UCU}@I_h(7)-C_{80}$ ' as following: '1.87 g of graphite powder and 1.83 g of U_3O_8 (1:24 molar ratio) are packed into each rod (6.7 g, without fillings). The collected raw soot was refluxed in chlorobenzene under Ar atmosphere for 12h and the solution was filtered. After solvent removal, the fullerenes were extracted by dissolving in toluene. On average ca. 140 mg of crude fullerene mixture per rod was obtained after arcing and extraction. In total, 60 carbon rods were burned and ca. 2 mg highly purified $\text{UCU}@C_{80}$ was obtained for characterization.' (Page 12, middle)

3. The authors report a ^{13}C NMR spectrum, which they assign to the carbon atoms of the $(I_h-C_{80})^{6-}$ moiety, while the bridging carbon atom was not observed. It would be nice to provide some comparisons from the literature to support this assignment. Is there any temperature dependence in the chemical shifts, since the cluster contains two paramagnetic U(V) atoms. There should be some temperature dependence.

Reply: Thanks to the reviewer for this kind suggestions. We modified the NMR discussion and added information to support this assignment (Page 8). We also carried out ^{13}C NMR measurements at 298K and 283K, and indeed we observed a slight temperature dependence of the chemical shifts. We replaced Figure S4 with a new figure including two NMR spectra at different temperatures. The discussion about the temperature dependence of the chemical shifts was added in the NMR discussion section (Page 8).

4. In the SI the authors report on the electrochemistry of the title compound (Fig. S11) and compare these observations to other complexes (Table S3). However, the authors do not discuss anything of these or even attempt to do so. Some comments are actually necessary to put this into perspective.

Reply: Thanks to the reviewer for the suggestions. We added a discussion of the electrochemical results and compared these to other I_h-C_{80} caged cluster fullerenes in the main text, on Page 10.

Reviewer #2 (Remarks to the Author):

This manuscript reports some rather important results and indeed deserves publication in *Nat. Comm*, but it comes to me the manuscript was prepared in a hurry manner and the writing was not equal. Substantial revisions are necessary before accept.

Theoretically:

1. Although the LUCUL unit can simulate the encapsulation as discussed, the authors should give reason why they choose $(C_7H_7)^{3-}$ and $(I_3)^{3-}$?

Reply: The I_h-C_{80} fullerene has a 5-fold degenerate HOMO with 4 electrons, thus it can accept 6 electrons to achieve a closed shell. The C_{80}^{6-} cage then uses 2 aromatic rings with 3 π -pairs each to form 2×3 dative pair bonds to the 2 U-5f6d valence shells. In order to work with smaller ligands for the calculations, with

similar electronegativity and similar electron accepting abilities, we chose 2 C₇H₇ rings (each C₇H₇ has a 2-fold degenerate HOMO with 1 electron, so they can accept 2×3=6 electrons), and alternatively 2 I₃-groups (I is similarly electronegative as C, and each I can accept 1 electron and offer a pair-donation). This is now explained in the MS, see pages 6-7.

2. The optimized geometrical structures of UCU@I_h(7)-C₈₀ and LUCUL should be given in the manuscript.

Reply: The respective geometric parameters are now listed in Table S5 on page 21 seq. of the SI file.

3. The optimizations of the three complexes in Table 1 were carried out using CASSCF (10e, 12o) method? If yes, the authors should definitely point it out in the manuscript.

Reply: All complexes were geometrically optimized at various DF approximation levels and then single point calculations were performed at ab-initio correlated spin-orbit coupled levels. Details are explained in the MS on pages 6-7 & 13, and the SI file on pages S3-S5.

4. How to confirm the formal oxidation state as +5?

Reply: We apply the common, recently updated IUPAC definition: draw a Lewis structure compatible with the findings, then assign the bond pairs to the more electronegative elements. L≡U=C=U≡L yields L^(-III) U^(+V) C^(-IV) U^(+V) L^(-III). We added an explanation in the MS on page 6.

5. Fig. 3, the color of ELF seems to be ambiguous, for example, the blue color in different locations.

Reply: We have added a color-scale with numerical values to Fig. 3, page 19.

6. Page 7, the U-C Mayer bond orders seem to be much lower compared to the formal U-C double bonds.

Reply: We now also discuss the larger EBO values, see bottom of page 7.

7. All the Figures and Tables in Supplementary Materials should be discussed in the manuscript.

Reply: Thanks to the reviewer for this suggestion. We now comment about these materials and refer explicitly to all of them (i.e. supplementary Texts, Figures, Tables and References) either directly in the MS, or in the referenced Supplementary Text (Figs. S8, S9, S13-S16 and Tabs. S3-S4, and in particular the Supplementary References, are mentioned on pages S2-S5).

Experimentally:

1. The synthesis of complex 1 is quite complicated, so I would like to know how much it yields.

Reply: The synthetic method used in this work is a 'physical procedure' rather than a typical chemical synthetic process, thus the actual product yield is difficult to estimate and is extremely low (much less than 1%). But to address the reviewer's concern, we added more experimental details: '1.87 g of graphite powder and 1.83 g of U₃O₈ (1:24 molar ratio) are filled in each rod (6.7 g, without fillings). The collected raw soot

was refluxed in chlorobenzene under Ar atmosphere for 12 h and the solution was filtered. After solvent removal, the fullerenes were extracted by dissolving in toluene. On average ca. 140 mg of crude fullerene mixture per rod was obtained after arcing and extraction. In total, 60 carbon rods were burned and ca. 2 mg highly purified U₂C@C₈₀ was obtained for characterization.’ (Page 12, middle).

2. The molecular composition and structure of complex 1 were determined by MALDI-TOF/MS and single-crystal X-ray diffraction analysis. These characterization methods are not comprehensive. Elemental analysis should be included in the manuscript.

Reply: Thanks for the reviewer’s suggestion. However, elemental analysis is not very practical and seldom used to characterize endohedral fullerenes, mainly because of small sample sizes. Exact masses from mass spectroscopy also lead to the establishment of molecular formulas. Nevertheless, we also used an alternative method, TEM-EDS analysis, which can also give the information of elemental composition of the molecule. This is now mentioned in the MS on page 4 bottom, and the SI-file page S2 and Fig. S1a on page S6.

3. The purity of the isolated U₂C@C₈₀ was confirmed by MALDI-TOF/MS. However, the purity of complex 1 was not mentioned in the manuscript. The purity of the sample is the prerequisite for all characterization. Therefore, the PXRD patterns of complex 1 should be included to determine purity.

Reply: Thanks to the reviewer for the kind suggestions.

(1)First, we have to apologize that we might mislead the reviewer by misusing the ‘ complex 1 ’ referring to the samples we used for characterizations. In fact, complex 1 has been only used in the single crystal analysis and was directly obtained from slow diffusion of nickel(II) octaethylporphyrin (Ni^{II}-OEP) in benzene into a CS₂ solution of UCU@C₈₀, only as single crystals (for details please see Page 13, Single Crystal X-ray Diffraction Analysis). All the rest of characterizations we present in the manuscript are based on the pure sample of UCU@C₈₀, not the complex 1. We have corrected these mistakes in the text by replacing ‘ compound 1 ’ by ‘UCU@C₈₀’.

(2) The PXRD method may be a way to determine the purity of complex 1, but with the sample amount we currently have, it would be very difficult to get a well-defined pattern. The sample amount of complex 1 was extremely low (estimated less than 0.1mg). Thus, it is not really practical for PXRD analysis. In fact, the PXRD method was only used in the early studies of endohedral fullerenes when single crystals of the endohedral fullerenes could not be obtained. In recent times, single crystals of the endohedral fullerenes are routinely obtained, as in the current work, thus PXRD is not really necessary. The high quality of the single crystal data and the resulting fine crystal structure we obtained from the single crystal Complex 1 sample are solid proof of its high purity.

4. No signal was detected for the single bridging C₀ carbon atom in the ¹³C NMR spectrum of complex 1. The author should offer an explanation.

Reply: We added some explanation on page 8, middle, as following: ‘Unfortunately, no signal was detected for the single bridging C₀ carbon atom, likely due to the limited sample amount as well as the paramagnetic broadening effects from the unpaired 5f¹ electrons on each uranium and the expected large and unknown shift at the center of the aromatic cage’.

5. There are 11 figures in the SI, only 5 of them were mentioned in the manuscript. Is it means that the other 6 are not necessary? The author should delete these figures or add some words to describe them in the manuscript

Reply: Thanks to the reviewer for this suggestion. We now comment about all figures explicitly (i.e. supplementary Texts, Figures, Tables and References) either directly in the MS, or in the referenced Supplementary Texts (Figs. S13-S16 and Tabs. S3, S4, and in particular the Supplementary References, are mentioned on pages S2-S5)

Minor suggestions:

(1) The label “b” is lost in Table 1.

Reply: Thank you for noting this typo. The wrong label ^a at the number 2.33 has now been corrected to ^b.

(2) Most of the Supplementary Materials is not referred (Fig S6-S11, Table S3) in the manuscript and some is wrong number, such as “The calculated atomic partial charges of order ± 1 are listed in Supplementary Table S3” should be “.....Table S2”

Reply: Thanks to the reviewer for this suggestion. It has been corrected. See response to remark 5 on the previous page.

Reviewer #3 (Remarks to the Author):

This paper describes the realisation of an important synthetic result – the isolation of a molecular compound that contains unsupported U=C bonds. The compound consists of a diuranium carbide unit, U=C=U, encapsulated within a C₈₀ fullerene cage. Uranium-element multiple bonding is of immense topical interest as studies of these compounds can shed light on actinide covalency, which is of importance in actinide/lanthanide separations in nuclear fuel cycles. Furthermore, compounds containing U=E bonds can exhibit interesting reactivity profiles and are interesting for fundamental reasons.

The diuranium carbide compound in this paper is the first molecular example of such a motif, hence the journal requirements of originality, significance and impact have been met. This work would therefore be a welcome addition to the literature and it is an appropriately important result to be considered for publication in this journal. However there are some points for the authors to address before I can recommend publication:

1. Title: shouldn't contain the word “unprecedented” or “first”; this is a standard requirement as all novel results are unprecedented.

Reply: Thanks for the reviewer's kind suggestion. We have deleted ‘unprecedented’ from the title.

2. Introduction: the referencing is not wholly appropriate. Many references appear to be given to a somewhat random selection of primary research papers giving homage to leading names in the field, rather than a more methodical selection of appropriate reviews. For example, refs 5-7 on actinide-ligand multiple bonding lagging behind the transition metals contains one example of molecular U=P multiple bonding and two

examples of matrix-isolated $U=CH_2$. Whilst these three references should be cited at some point in the paper the correct references at this point are surely ref #8 and Hayton's other $U=E$ review (Dalton Trans., 2010, 39, 1145) should also be added here. Then, when stating that unsupported $U=C$ bonds are conspicuously missing the most appropriate review by Liddle is currently missing (Comments Inorg. Chem., 2015, 35:5, 262; Liddle has written three other reviews on uranium carbenes that could also be cited, plus his comprehensive review on molecular uranium chemistry is also appropriate: Angew. Chem. Int. Ed., 2016, 54, 2). Quite why Mazzanti's excellent work on uranium nitride functionalisation is referenced at this point (#12) is unclear. The authors should state what they mean by "unsupported", as the Hayton and Gilje ylidic examples in the next sentence are not pincer complexes, they are merely phosphorus-stabilised.

Other appropriate omissions from the introduction that require inclusion include: $U=Se$, $=Te$ and $=As$ bonds have also been isolated, with appropriate Hayton/Liddle references (bottom of page 2); a clear statement that $U=C$ bonds have been observed by matrix isolation (top of page 3); the first uranyl(VI) carbene (Ephritikhine, JACS, 2011, 133, 6162), the first uranium(V) carbene (Liddle, Angew. Chem. Int. Ed., 2011, 50, 2383) and a reference to Gilje for the $Cp_3U=C$ system (JACS, 1981, 103, 3589) (all top of page 3); a reference should be added to uranium carbide ceramics (middle of page 3). Finally, as this is a molecular version of uranium carbide, which has been known for over a hundred years and is of relevance to nuclear cycles, the authors should make some reference to this work for context. I am unfamiliar with this area but the review by Frad (Adv. Inorg. Chem. Radiochem., 1968, 11, 153) appears to be a good place to start.

For example, refs 5-7 on actinide-ligand multiple bonding lagging behind the transition metals contains one example of molecular $U=P$ multiple bonding and two examples of matrix-isolated $U=CH_2$. Whilst these three references should be cited at some point in the paper the correct references at this point are surely ref #8 and Hayton's other $U=E$ review (Dalton Trans., 2010, 39, 1145) should also be added here.

Then, when stating that unsupported $U=C$ bonds are conspicuously missing the most appropriate review by Stephen T. Liddle is currently missing (Comments Inorg. Chem., 2015, 35:5, 262; Liddle has written three other reviews on uranium carbenes that could also be cited, plus his comprehensive review on molecular uranium chemistry is also appropriate: Angew. Chem. Int. Ed., 2015, 54, 8604-8641, The Renaissance of Non-Aqueous Uranium Chemistry).

Other appropriate omissions from the introduction that require inclusion include: $U=Se$, $=Te$ and $=As$ bonds have also been isolated, with appropriate Hayton/Liddle references (bottom of page 2); a clear statement that $U=C$ bonds have been observed by matrix isolation (top of page 3); the first uranyl(VI) carbene (Ephritikhine, JACS, 2011, 133, 6162), the first uranium(V) carbene (Liddle, Angew. Chem. Int. Ed., 2011, 50, 2383) and a reference to Gilje for the $Cp_3U=C$ system (JACS, 1981, 103, 3589) (all top of page 3); A reference should be added to uranium carbide ceramics (middle of page 3).

Finally, as this is a molecular version of uranium carbide, which has been known for over a hundred years and is of relevance to nuclear cycles, the authors should make some reference to this work for context. I am unfamiliar with this area but the review by Frad (Adv. Inorg. Chem. Radiochem., 1968, 11, 153) appears to be a good place to start.

Quite why Mazzanti's excellent work on uranium nitride functionalisation is referenced at this point (#12) is unclear. The authors should state what they mean by "unsupported", as the Hayton and Gilje ylidic examples in the next sentence are not pincer complexes, they are merely phosphorus-stabilised.

Reply: We thank the reviewer's very much for his kind and very detailed suggestions.

(1) Accordingly, we have reorganized the introduction and added various reference papers. We have now expanded the number of references in the introduction from 17 to 41, which include in particular those

suggested by the reviewer. For details please see revised introduction on pages 2-3 and references 1-44.

(2) We have added more text at the bottom of page 2 to better describe the ‘unsupported U=C’ bond: ‘Conspicuously missing from this list are U=C bonds, specifically those which lack ancillary heteroatom or chelating support. As metal-carbon double bonds such as these have become commonplace in transition metal chemistry and catalysis, unsupported U=C bonds have long remained a major, outstanding synthetic target.’

(3) We have also added in the middle of page 3 the following sentence: ‘The knowledge gained from studying stable compounds featuring U=C multiple-bonds can be extended to uranium carbide ceramics, which offer improved thermal density and higher conductivity over current UO₂ nuclear fuels.’ to address this background knowledge.

3. In the last sentence of the introduction change “first example” to “first structurally characterised example” (to acknowledge matrix isolation studies). Esds should also be added to the bond lengths and angles in this last paragraph of the intro and paragraphs 1 and 2 of page 5.

Reply:

(1) Thanks to the reviewer for the kind suggestion. We have changed “first example” to “first structurally characterized example” at the bottom of page 3.

(2) We have added the Esds in the introduction (top of page 4) and in the section on structure (pages 4-6).

4. **Molecular structure discussion**: The U-arene distances are compared to those in an example by Diaconescu that contains a formally dianionic C₇H₈ unit and U(III) centres. As the authors claim U(V) centres they could also compare these distances to the sole example by Liddle (Angew. Chem. Int. Ed., 2011, 50, 10388) or terminal U-arene complexes rather than inverse sandwiches (e.g. Inorg. Chim. Acta., 1971, 5:3, 439 and Cotton examples in the 1980s). The Liddle reference no. in paragraph 2 on page 5 should be #13, not #14. Finally, Boncella’s RN=U=NR uranyl analogue complex (Science, 2005, 310, 1941) should be referenced too at the bottom of page 5 as these bond lengths (1.844(6) Å) are also shorter than the U=C bonds herein.

Reply: Thanks to the reviewer, we have now more correctly referred to multiple internuclear U-C distances in the literature (Liddle (Angew. Chem. Int. Ed., 2011, 50, 10388) has been cited as ref 48 in Page 5) and have pointed out five aspects concerning the scattering: (i) the competition of axial and equatorial bonding, concerning shorter and stronger vs. longer and weaker bonds, (ii) the tendency that electronegative elements such as O and N in general preferring strong and short axial bonding more than C and B do, (iii) the bond lengths decrease with increasing oxidation state of U; (iv) the bond lengths decrease with decreasing coordination number; (v) the complication by multidentate ligands and spatially demanding ligands. See the manuscript, pages 2, 5, 7, 11.

5. Spectroscopic properties: the chemical shift range for the ¹³C NMR experiment should be provided; U(V) complexes can give unusual NMR shifts so it is important for the authors to state this. Whilst I am unsurprised that no signal was observed it could be as deshielded as 500 ppm.

Reply: Thanks for the reviewer’s suggestion. We have discussed the chemical shift of the ¹³C NMR experiment (-15 to +250 ppm) on p. 8 of the MS and added some experimental details in the SI, see pages

S2 & S9. Unfortunately this does not cover the possibility mentioned by the reviewer, of a deshielding of 500 ppm.

5a. Furthermore, as the authors have concluded **U(V) centres** are present they should perform standard characterisation techniques to **confirm this** and show the highest levels of rigour, which can include: (i) **NIR** spectroscopy. The spectrum currently provided in the SI has not been converted to give extinction coefficients; could the authors provide concentrations as concentrated U(V) solutions often **give absorbances** at ca. 6700 cm^{-1} ($\sim 1250\text{ nm}$) for an f-f transition; (ii) VT DC measurements from 300 K - 2 K using a SQUID can **confirm the magnetic 5f1** U(V) ground state as even at 2 K there will be a significant magnetic susceptibility; (iii) **EPR spectroscopy** can also confirm the 5f1 configuration; 5-10 K should be the ideal temperature to carry these experiments out.

Reply: (i) **NIR spectroscopy:** The values of 6700 cm^{-1} ($\sim 0.83\text{ eV}$) and 1250 nm ($\sim 0.99\text{ eV}$) mentioned by the reviewer indeed correspond to the 0.76 eV excitation gap calculated for $L>U(f^1)=C=U(f^1)<L$ in Table S4. These Laporte forbidden f->f transitions are very weak. In general, we need concentrations of 10mM or greater to observe them. However, endohedral fullerenes normally have pretty low solubilities. According to the reviewer's suggestion, we have tried to increase the concentration of UCU@C₈₀ in CS₂ and have measured the NIR spectrum again, but **we didn't observe any absorption peaks down to 0.775 eV (1600 nm)**, see Fig. S5a. If there are some features a bit lower, we could not see them with our equipment.

(2) VT DC measurements from 300 K - 2 K using a SQUID: We have performed such measurements and found a Curie-type paramagnetism visible below say 40K (on top of some huge temperature independent magnetism) corresponding to a molecular μ_{eff} around $1\ \mu_{\text{Bohr}}$, see MS page 11 and Fig. S9 and text ST5. According to our calculation, the magnetism of the LU(f¹)CU(f¹)L unit is due to the two spins (with mixed singlet-triplet coupling) and the angular momenta of the dominantly contributing U-5f_{±3} orbitals, partially quenched due to strong SO coupling and weak orbital coupling. The lower states are severely configuration mixed and near degenerate, so that there is a thermally variable population. Our CAS-PT2 SO-SI calculations are not accurate enough to obtain really reliable results at the low temperatures. For the case where the C₈₀ ligand of UCU was replaced by the smaller C₁₄H₁₄, we obtained μ_{eff} values of 3-4 μ_{Bohr} .

(3) EPR spectroscopy: We performed measurements at 5K, as described in the MS (page 11) and the SI (pages S3 and S11), but only obtained a completely washed out feature.

At present, we tried our best but cannot improve the situation with the very limited sample amount. We intend to synthesize more substance and then perform further magnetic measurements in the future.

6. Conclusions: the authors state that they have discovered **"genuine" U=C bonding**. I think that as written this sounds a little disparaging of previous work (why are the matrix isolation experiments not genuine, or the phosphorus-stabilised work?), so should be rephrased as this work is complementary. Furthermore, this is not **an "elusive" uranium-element multiple bond**, as the authors have currently stated.

Reply: We have replaced **"genuine"** with **"unsupported"** and **deleted the word 'elusive'**. Further, we have **rewritten** part of the paragraph, please see top of page 12 of the revised MS.

7. Methods: Although I expect that the yields of this product are extremely low this has not currently been stated and there is currently not enough detail for another group to repeat this work. Therefore, the authors should add **how much U₃O₈ and graphite powder** was used in grams, together with the **crude yield** in g, and

the crystalline yield in g and mmol. I imagine that there is a sub-1% yield based on U_3O_8 , so no need for percentage yields if so.

Reply: Thanks for the reviewer's suggestion. See similar comments (with detailed responses) by the two other reviewers on pages 2-top and 4-middle above. The yield of the final purified product is indeed extremely low. Additionally, the synthetic method is a physical procedure rather than a typical chemical synthetic process, which also makes it difficult to state the exact product yield. But we agree with the reviewer that more details are needed for another group to repeat this work and we added the following experimental details into the Methods section: '1.87 g of graphite powder and 1.83 g of U_3O_8 (1:24 molar ratio) are filled in each rod (6.7 g, without fillings). The collected raw soot was refluxed in chlorobenzene under Ar atmosphere for 12 h and the solution was filtered. After solvent removal, the fullerenes were extracted by dissolving in toluene. On average ca. 140 mg of crude fullerene mixture per rod was obtained after arcing and extraction. In total, 60 carbon rods were burned and ca. 2 mg highly purified $U_2C@C_{80}$ was obtained for characterization.' (Page 12, middle). We also gave the experimental details for the preparation of co-crystals of $UCU@I_h-C_{80} \cdot [Ni^{II}-OEP]$ as following : Black co-crystals of $UCU@I_h-C_{80} \cdot [Ni^{II}-OEP]$ were obtained by allowing the benzene solution of $[Ni^{II}-OEP]$ (1.2 mg/mL, 0.8 mL) and the CS_2 solution of $UCU@I_h-C_{80}$ (1 mg/mL, 0.5 mL) to diffuse together.

Reviewers' comments:

Reviewer #1 (Remarks to the Author):

The authors have responded well to the criticism/suggestions raised by the referees. So most issues have been resolved with the exception of the magnetism reported by the authors which is certainly troublesome and I would strongly urge the authors to repeat this measurement. Otherwise, I don't see a chance (at least from my perspective) that this manuscript should be published. The results are certainly of great interest, but the magnetic data doesn't make much sense. The authors state in the Supporting Information that their sample weight varies between 0.3-0.4 mg (i.e. this corresponds to an error of ca. 30%). Furthermore, no comment was made if the magnetic data was corrected for the diamagnetism of the sample (i.e. Pascal constants). The authors also state that because of the experimental error the values presented on Figure S9 should be considered as "a.u.", yet the figures show the physically correct units. A value of $\chi T = 38 \text{ cm}^3 \text{ mol}^{-1} \text{ K}$ would correspond according to $\mu(\text{eff}) = (8 \chi T)^{0.5} = 17.4 \mu\text{B}$, which is completely out of range of what would be expected. Have field sweeps be performed to obtain any information on the identity of a potential impurity? In the end only reliable and reproducible data should be published.

Reviewer #2 (Remarks to the Author):

The authors have decently replied all of my concerns, now I have to recommend publication of this manuscript in Nat. Commun.

Reviewer #3 (Remarks to the Author):

The authors have done a commendable task in addressing referees comments by carrying out further experiments and making requested changes to the manuscript. However, a couple of points stand out that the authors still need to do to address these original comments in full:

1. Despite undertaking the requested SQUID, EPR and NIR spectroscopy experiments the U(V) formulation has not been confirmed as the data can be interpreted as $2 \times \text{U(V)}$ or a mixed U(IV)/U(VI) system, or even a system with unpaired e⁻ delocalised about the framework. It is quite acceptable for the authors to postulate and assign +5 oxidation states based on first principles but this has not been confirmed experimentally; thus all discussion in the paper should be changed to make this clear – this includes the abstract, which currently states U(V) centres as if it is a fact, and the discussion, which states that pentavalent uranium is "conclusive" based on experimental data (it is not as the three most appropriate methods of proving this are not unambiguous in this case).

2. Reviewer 2 suggested a PXRD experiment to determine bulk purity of complex 1, and the authors state that they don't have enough for PXRD analysis. In the rebuttal the authors state that the complex is pure by single crystal XRD, but this is not an acceptable method for analysing bulk purity, which is what the reviewer wanted (a clear advantage of PXRD here is to show bulk sample homogeneity). There appears to be some confusion on both sides here about what needed to be analysed, so can the authors please carry out PXRD on UCU@C80 to see if there is only one phase to make some effort to address this comment. The authors could even alternatively "crash-crystallise" a larger amount of complex 1, as crystals obviously do not need to be single for PXRD.

Response to Reviewers' comments:

According to the reviewer's comments, we made the following changes in the manuscript.

Reviewer #1 (Remarks to the Author):

The authors have responded well to the criticism/suggestions raised by the referees. So most issues have been resolved with the exception of the **magnetism** reported by the authors which is certainly troublesome and I would strongly urge the authors to **repeat this measurement**. Otherwise, I don't see a chance (at least from my perspective) that this manuscript should be published. The results are certainly of great interest, but the magnetic data doesn't make much sense. The authors state in the Supporting Information that their sample weight varies between 0.3-0.4 mg (i.e. this corresponds to an error of ca. 30%). Furthermore, no comment was made if the magnetic data was **corrected for the diamagnetism** of the sample (i.e. Pascal constants). The authors also state that because of the experimental error the values presented on Figure S9 should be considered as "a.u.", yet the figures show the physically correct units. A value of $\chi T = 38 \text{ cm}^3 \text{ mol}^{-1} \text{ K}$ would correspond according to **$\mu(\text{eff}) = (8 \chi T)^{0.5} = 17.4 \text{ } \mu\text{B}$** , which is completely out of range of what would should be expected. Have **field sweeps** be performed to obtain any information on the identity of a potential impurity? In the end only reliable and reproducible data should be published.

Responses by the authors

Thanks to this reviewer for the comments and kind suggestions:

(1) We have **repeated** the magnetic measurements at the IFW Dresden, (Germany) with the help of Dr. Popov, and used the new data in this revised version. (see MS p.11/12 and SI p.3/12.) Only 0.1 mg substance was used for these measurements, but the present results replicated the pre-

vious findings obtained from the ICCAS in Beijing(China) at 0.1 Tesla. Both measurements showed a similar low-temperature Curie-type temperature-dependent magnetism. We admit that our absolute values of χ and μ have significant absolute errors, but the temperature and field dependencies are very consistent, and also with those of the theoretical simulations.

(2) In particular, the new measurements were performed at **0.2 up to 7 Tesla** and agree well with each other. They are shown in supplementary Fig. 9-left. Altogether, the magnetic data appear quite reliable.

(3) We corrected for the expected diamagnetism in the previous measurements and have done so again for the new measurements (the encapsulation polymer, etc). The dominant background comes from temperature independent paramagnetism, and we also tried to correct for both measurements. (For details please see Supplementary material – p.3, Method 5: Magnetometric Measurements; p.12, Figure 9: Magnetism $T \cdot \chi$ of UCU@C80 vs. T (in K) in various magnetic fields; see also Manuscript – p.12.)

(4) We stressed before, and again we do so here: The molecular effective **magnetic moments $\mu(\text{eff})$** causing the temperature-dependent Curie-type paramagnetism are on the order of $1 \pm \mu_{\text{Bohr}}$. We stress that it makes little physical sense to derive a Curie-type effective moment from temperature-independent paramagnetism and then discussing the purely formal result of $\mu_{\text{eff}} \sim \sqrt{T}$.

One final comment concerning the magnetic properties of this compound: While potentially very interesting, it is not the central focus of the present work which is mainly centred on the usual structural and electronic properties of this unique compound. As the current magnetic data has been proved to be reliable, we hope that it is acceptable for the final publication. As we obtain larger quantities of this compound, we will readdress the magnetism properties in the future.

Reviewer #3 (Remarks to the Author).

Despite undertaking the requested **SQUID, EPR and NIR** spectroscopy experiments, the U(V) formulation has **not been confirmed** as the data can be interpreted as **2 x U(V) or a mixed U(IV)/U(VI) system, or even a system with unpaired e- delocalised** about the framework. It is quite acceptable for the authors to postulate and assign +5 oxidation states based on first principles but this has **not been confirmed experimentally**; thus all discussion in the paper should be changed to make this clear – this includes the abstract, which currently states U(V) centres as if it is a fact, and the discussion, which states that pentavalent uranium is “conclusive” based on experimental data (it is not as the three most appropriate methods of proving this are not unambiguous in this case).

Responses by the authors

Thanks to this reviewer for the kind suggestions.

(1) We had thought about the three possibilities: (i) $U(f^1) \rightarrow U(f^1)$, (ii) $U(f^2) \rightarrow U(f^0)$, (iii) $U_2(f^x) \rightarrow C_{80}(f^y)$. We list our arguments, pointing toward case (i).

(2) The experimental X-ray diffraction results are compatible with symmetric, not with non-symmetric U-C-U, pointing against (ii).

(3) The measured Uranium inner-shell ionization shows features (supplementary Fig. 7) of only one U species, also pointing against (ii).

(4) The SQUID measurements can be simulated by a weakly coupled pair of two U(V), giving support for option (i).

(5) The density functional calculations of $U_2C@C_{80}$ and of the model molecules $U_2C@(C_7H_7)_2$ and $U_2C@(I_3)_2$, as well as the CASSCF calculations of the $U_2C@(C_7H_7)_2$ model all support symmetric U(V)--U(V) against U(IV)—U(VI).

(6) In particular, the experimental and U(V)--U(V)-calculated IR spectra agree reasonably.

(7) The expected f-f NIR features were not found. That may be due to low intensity and small amount of substance, or that our equipment could only measure down to ca. 0.8 eV.

(8) Whether there is unpaired electron and spin density on the fullerene cage, is indeed an open question. The density functional calculations indicated very little such density. On the other hand, the extremely broad EPR signal and the huge temperature-independent paramagnetism request for further investigation, as we mentioned in the manuscript.

In summary, the presumption of a U(V)—U(V) compound appears to be reasonably well founded. Following reviewer's suggestion, we revised the phrase in the abstract and the conclusion since there are indeed open questions (EPR and SQUID) to be further investigated in future studies, for details please see MS page1 and page12.

Reviewer #3 (Remarks to the Author):

Reviewer suggests a PXRD experiment to determine bulk purity of complex 1, and the authors state that they don't have enough for PXRD analysis. In the rebuttal the authors state that the complex is pure by single crystal XRD, but this is not an acceptable method for analysing bulk purity, which is what the reviewer wanted (a clear advantage of PXRD here is to show bulk sample homogeneity). There appears to be some confusion on both sides here about what needed to be analysed, so can the authors **please carry out PXRD on UCU@C80** to see if there is only one phase to make some effort to address this comment. The authors could even alternatively "crash-crystallise" a larger amount of complex 1, as crystals obviously do not need to be single for PXRD. The purity of the isolated U₂C@C₈₀ was confirmed by MALDI-TOF/MS. However, the purity of complex 1 was not mentioned in the manuscript. The purity of the sample is the prerequisite for all characterization. Therefore, the PXRD patterns of complex 1 should be included to determine purity.

Responses by the authors

Thanks to this reviewer for the kind suggestions. We have now performed the requested powder XRD investigation of UCU@C₈₀. It corroborates the phase-purity of UCU@C₈₀ used for experimental characterizations. This is now mentioned in the MS on page 6/14, the spectrum is shown in the supplementary file on page 9.

REVIEWERS' COMMENTS:

Reviewer #1 (Remarks to the Author):

The authors addressed my original concerns and therefore I believe the paper is acceptable.

However, I have two more comments for the authors to consider:

Although I am personally skeptical with respect to the magnetic data and their TIP background correction, the magnetic data now reported is much more in line with a reasonable expectation value, so that I don't want to hold up acceptance of the manuscript. Nevertheless, their comment "We stress that it makes little physical sense to derive a Curie-type effective moment from temperature-independent paramagnetism and then discussing the purely formal result of $\mu_{\text{eff}} \sim \sqrt{T}$." is incorrect. They previously mentioned a χT value of $38 \text{ cm}^3 \text{ mol}^{-1} \text{ K}$, which is physical non-sense. They authors explicitly state that the antiferromagnetic exchange coupling is small, so that one would expect two uncorrelated U(V, f1) centers, which can never reach a value of $\chi T = 38 \text{ cm}^3 \text{ mol}^{-1} \text{ K}$! Furthermore, their new χT value can now be compared to other papers reporting bimetallic U(V) systems for which antiferromagnetic coupling was previously observed (Edelstein, Andersen: JACS J. Am. Chem. Soc., 1990, 112, 4588–4590; Boncella: ACIE 2009, 48, 3795-3798)) should explicitly be included in the manuscript!!

"No clearly defined signal was not observed, see Supplementary Figure 8. Factors such as nuclear quadrupole couplings, near degenerate electronic states and cage shielding effects complicate the analysis." - This statement is incorrect, since the double negative "no clearly defined signal" "was not observed" implies that the authors DID observe a signal.

Reviewer #3 (Remarks to the Author):

I thank the authors for responding to the remaining comments from the reviewers. I have some further comments on their discussions: I do not suggest any further experiments. If the authors do these very minor changes then I do not need to see the manuscript again.

For referee #1, the authors have performed the requested duplicate measurement of the magnetic data, and have also collected the requested field-swept data. I agree with the authors that the magnetic data is not a focal point for the paper, so the consistent duplicate data collected appears sufficient as does most of the qualitative description in the manuscript.

However, in response point (1) the authors state that despite the significant absolute errors, the temperature and field dependencies are consistent and in line with theoretical simulations. But the authors can only use 0.1 mg - what is the accuracy of the scale that they used, and therefore what is the error? It could again be as high as 30%, so numerical values (and by extrapolation their comparison to calculated values) are somewhat meaningless. The authors can easily address this by stating this imprecision in the relevant Figure captions of the ESI, along with a comment that these are not absolute values but can be considered as a.u. due to this imprecision, as the referee stated.

Also, the huge TIP swamps the signal of interest, which makes the discussion of the data in the manuscript the wrong way around – the statement about TIP should be at the start of the paragraph, as this is the major component, and discussion of the finer detail at low temperature should be afterwards. This would correctly contextualise the limited interpretation of the data that can be done.

For referee #3, I am happy that the changes made to the presentation of the assignment of U(V) centres in the manuscript now match very well the interpretation that can be made of the data. I agree with the authors' logic and points in the response, which are not in the manuscript (so do

not require another rebuttal or any changes to the paper). I just present some counter-arguments for consideration to explain why caution has been urged from my part:

(2) When two non-identical sites are disordered almost equally over two positions in single crystal XRD they can appear symmetric (e.g. see Meyer's bridging CO U complex, or Liddle's latest U arene wheel).

(4) The SQUID data is swamped by TIP and has huge errors (see above and point 8).

(5) and (6) calculations are not experimental data.

(7) The lack of NIR/EPR signals can somewhat counter a U(V) argument, so there is as much experimental data for and against this formulation.

RESPONSE TO THE REVIEWERS' COMMENTS

Reviewer #1 (Remarks to the Author):

The authors addressed my original concerns and therefore *I believe the paper is acceptable. However, I have two more comments for the authors to consider:*

Although I am personally skeptical with respect to the magnetic data and their TIP background correction, the magnetic data now reported is much more in line with a reasonable expectation value, so that I don't want to hold up acceptance of the manuscript. Nevertheless, their comment "We stress that it makes little physical sense to derive a Curie-type effective moment from temperature-independent paramagnetism and then discussing the purely formal result of $\mu_{\text{eff}} \sim \sqrt{T}$." is incorrect.

RESPONSE: The magnetic susceptibility vs. T curves (similar observations from three different laboratories) exhibit two regimes, (i) one at 'high temperatures' from 300 K down to ca. 60 K, and (ii) another one at 'low temperatures' from ca. 40 K down to 2 K. This is a rather common feature in the field of uranium complex magnetism. Region (i) however is dominated in the present case by a temperature-independent magnetism (TIP), which consists of the expected diamagnetism and a much larger, novel huge paramagnetism. We so far have never read the suggestion to physically interpret a basically constant magnetism (be it Langevin/Pascal or Landau type diamagnetism or Pauli type paramagnetism) by thermally liberating molecular moments of internal square-root(T) variation. However, we admit that the subtraction of a basically constant background from the 'low temperature' magnetism may introduce significant uncertainty of the derived μ_{eff} , as also noted by reviewer #3. Therefore we had named the magnetic units 'arbitrary units' as a precaution and add corresponding lines in the figure caption as suggested by reviewer #3. At the moment, we cannot do more. However we intend to investigate the 'unprecedented huge TIP' in the future.

They previously mentioned a χT value of $38 \text{ cm}^3 \text{ mol}^{-1} \text{ K}$, which is physical non-sense. The authors explicitly state that the antiferromagnetic exchange coupling is small, so that one would expect two uncorrelated U(V, f1) centers, which can never reach a value of $\chi T = 38 \text{ cm}^3 \text{ mol}^{-1} \text{ K}$!

RESPONSE: When plotting the observed magnetism as χT vs. T , one indeed obtains a straight line with large slope, indicating the mentioned huge TIP of bigger than $+0.1 \text{ cm}^3/\text{mol}$. We were also quite confused. Therefore we asked 3 different laboratories to measure it, and reconfirmed it. We will research this unexpected observation in the near future, but would still keep our original speculative explanation in the manuscript that the huge TIP "may possibly be due to cooperative coupling of UCU and C_{80} in the solid and related to the low band-gap calculated for UCU@ C_{80} " (page 11 bottom).

Furthermore, their new χT value can now be compared to other papers reporting bimetallic U(V) systems for which antiferromagnetic coupling was previously observed (Edelstein, Andersen: JACS J. Am. Chem. Soc., 1990, 112, 4588–4590; Boncella: ACIE 2009, 48, 3795-3798)) should **explicitly be included** in the manuscript!!

RESPONSE: Thanks for the reviewer's suggestion. Our new 'low temperature' χT values corrected for the TIP background, are very similar to the previous 'low temperature' χT values corrected for the TIP background. As before, they correspond to Curie μ_{eff} values below $1 \mu_{\text{Bohr}}$ above $T = 2 \text{ K}$, and to μ_{eff} values below $2 \mu_{\text{Bohr}}$ below $T = 40 \text{ K}$. Such an increase of μ_{eff} is quite common for uranium complexes of

oxidation states V and IV. **We have now mentioned that explicitly in the manuscript (page 12 top), citing the two suggested references (nos. 64 and 65) as well as previous ones (refs.21, 22, 48) and further ones (refs. 66, 67, 68).**

“No clearly defined signal was **not** observed, see Supplementary Figure 8. Factors such as nuclear quadrupole couplings, near degenerate electronic states and cage shielding effects complicate the analysis.” - This statement is incorrect, since the double negative “no clearly defined signal” “was not observed” implies that the authors DID observe a signal.

RESPONSE: The authors apologize for their carelessness during the previous revision. **We have corrected this typo now and the present writing is ‘No clearly defined signal was observed, see Supplementary Fig. 8’.**

Reviewer #3 (Remarks to the Author):

I thank the authors for responding to the remaining comments from the reviewers. I have some further comments on their discussions: I do not suggest any further experiments. *If the authors do these very minor changes then I do not need to see the manuscript again.*

For referee #1, the authors have performed the requested duplicate measurement of the magnetic data, and have also collected the requested field-swept data. I agree with the authors that the magnetic data is not a focal point for the paper, so the consistent duplicate data collection appears sufficient as does most of the qualitative description in the manuscript.

(1) However, in response to point (1) the authors state that despite the significant absolute errors, the temperature and field dependencies are consistent and in line with theoretical simulations. But the authors can only use 0.1 mg - what is the accuracy of the scale that they used, and therefore what is the error? It could again be as high as 30%, so numerical values (and by extrapolation their comparison to calculated values) are somewhat meaningless. The authors can easily address this by stating this imprecision in the relevant Figure captions of the ESI, along with a comment that these are not absolute values but can be considered as a.u. due to this imprecision, as the referee stated.

RESPONSE: The Reviewer is right that the deduced magnetic μ_{eff} values are contaminated by large error bars, both due to the small amount of substance, and due to the uncertainty of the background subtraction. Therefore we are calling the observed units ‘arbitrary’ although meaning $\text{cm}^3\text{mol}^{-1}$, but 30% error may be possible. **We have explicitly noted that now by adding ‘Due to the hypothetical background-correction and the small amount of substance, the absolute values for χT have large uncertainties, therefore the $\text{cm}^3\text{mole}^{-1}\text{K}$ units are named ‘arbitrary units’ as a precaution’ in the caption of Supplementary Figure 9 as suggested by the reviewer.**

Also, the huge TIP swamps the signal of interest, which makes the discussion of the data in the manuscript the wrong way around – the **statement about TIP should be at the start** of the paragraph, as this is the major component, and discussion of the finer detail at low temperature should be afterwards. This would correctly contextualize the limited interpretation of the data that can be done.

RESPONSE: **We have followed the suggestion of the reviewer and rearranged the sentences, see lower part of page 11 of the manuscript. We thank the reviewer for suggesting this improvement.**

For referee #3, I am happy that the changes made to the presentation of the assignment of U(V) centres in the manuscript now match very well the interpretation that can be made of the data. I agree with the authors' logic and points in the response, which are not in the manuscript (so do not require another rebuttal or any changes to the paper). I just present some counter-arguments for consideration to explain why caution has been urged from my part:

RESPONSE: We understand that. However, after finding so many evidences not in clear disagreement with our interpretation, we finally felt, that our interpretation has a high degree of probability.

(2) When two non-identical sites are disordered almost equally over two positions in single crystal XRD they can appear symmetric (e.g. see Meyer's bridging CO U complex, or Liddle's latest U arene wheel).

RESPONSE: However, when doing the structure analysis, and searching for various partially occupied U sites, we found them in the transversal direction, not in the bond-parallel direction.

(4) The SQUID data is swamped by TIP and has huge errors (see above and point 8).

RESPONSE: We have always admitted that. And we admit: we can only say that the experimental data are not in disagreement with the theoretical model. The fitting simulation is at least not against our interpretation.

(5) and (6) calculations are not experimental data.

RESPONSE: We agree. And again no contradiction to our interpretation.

(7) The lack of NIR/EPR signals can somewhat counter a U(V) argument, so there is as much experimental data for and against this formulation.

RESPONSE: We admit that we didn't get support from these data. This lack may be related to the huge TIP, which we will continue to investigate in the near future.